# Structural basis for DNA proofreading

Gina Buchel [1,2], Ashok R. Nayak[1,2], Karl Herbine [1], Azadeh Sarfallah[1], Viktoriia O. Sokolova[1], Angelica Zamudio-Ochoa[1] & Dmitry Temiakov [1] ✉

DNA polymerase (DNAP) can correct errors in DNA during replication by proofreading, a process critical for cell viability. However, the mechanism by which an erroneously incorporated base translocates from the polymerase to the exonuclease site and the corrected DNA terminus returns has remained elusive. Here, we present an ensemble of nine high-resolution structures representing human mitochondrial DNA polymerase Gamma, Polγ, captured during consecutive proofreading steps. The structures reveal key events, including mismatched base recognition, its dissociation from the polymerase site, forward translocation of DNAP, alterations in DNA trajectory, repositioning and refolding of elements for primer separation, DNAP backtracking, and displacement of the mismatched base into the exonuclease site. Altogether, our findings suggest a conserved 'bolt-action' mechanism of proofreading based on iterative cycles of DNAP translocation without dissociation from the DNA, facilitating primer transfer between catalytic sites. Functional assays and mutagenesis corroborate this mechanism, connecting pathogenic mutations to crucial structural elements in proofreading steps.

Maintaining fidelity of replication of genetic information is among the most critical functions of living organisms. Errors arise as a result of DNA damage but also owing to the occasional incorporation of incorrect (non-cognate) substrates, resulting in mismatched base pairs and potentially deleterious mutations[1]. Cells have evolved sophisticated mechanisms to fix these errors[2]; among them is the ability of DNAP to correct the mismatched bases during DNA replication by a proofreading activity[3–5]. Proofreading employs an intrinsic exonucleolytic activity present in DNAP, or the exonucleolytic activity of auxiliary factors[6,7]. The former is observed in the Pol A family of DNAPs, which includes bacterial DNA Polymerase I, bacteriophage T7 DNAP, and human mitochondrial DNAP Polγ[4,8–12]. The N-terminus of these polymerases harbors an exonuclease domain capable of excision of a terminal nucleotide in a canonical metal ion-dependent reaction[13]. The exonuclease (*exo)* site is located ~35 Å away from the polymerase (*pol) site*, with no direct path between them. Therefore, it remains unclear how the misincorporated nucleotide in the nascent DNA can be transferred into the *exo* site, what triggers the primer separation from the template strand, and how the corrected terminus returns to the *pol* site after the cleavage[6,14,15]. One of the hypotheses is based on an intermolecular model, which postulates that upon incorporating a non-cognate base, DNAP dissociates and rebinds the mismatched primer in the *exo* site[16,17]. Alternative models suggest an intramolecular mode of proofreading, during which polymerase stays associated with the DNA, but the primer terminus shifts from the *pol* site to the *exo* site and returns[18–21] or a combination of the two modes[14]. While there is a general agreement in the field that Pol A enzymes are processive and can proofread DNA without dissociating, the lack of understanding of how editing can be achieved without the engagement of dedicated structural elements in DNAP has persisted until now[6,14,15]. In this study, we provide a structural basis for the mechanism of proofreading, termed here as "bolt-action," by human mitochondrial DNAP Polγ and describe the major steps of this process.

## Results and discussion
### Capturing Polγ during the multi-step proofreading process
To study the proofreading mechanism, we hypothesized that if a processive DNAP, such as Polγ, does not dissociate from the DNA template, it must traverse the misincorporated base from the *pol* to the *exo* site through intermediate steps. Two strategies were used to capture these intermediates by preventing the exonucleolytic activity of the enzyme. In the first, we used Wild Type (WT) Polγ (Fig. 1a) and a

[1]Department of Biochemistry and Molecular Biology, Thomas Jefferson University, 1020 Locust St, Philadelphia, PA 19107, USA. [2]These authors contributed equally: Gina Buchel, Ashok R. Nayak. ✉e-mail: dmitry.temiakov@jefferson.edu

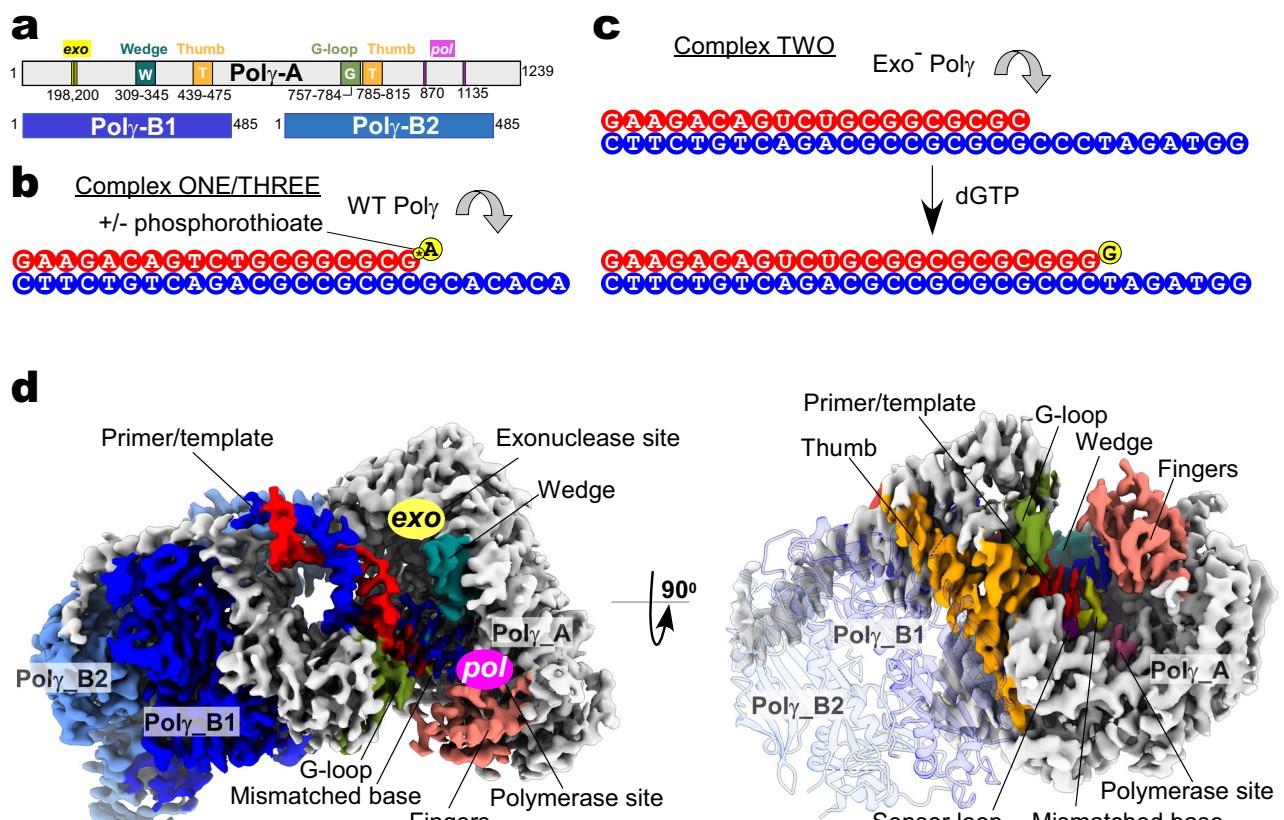

**Fig. 1 | CryoEM structure of Polγ. a** A schematic showing the domain organization of the Polγ holoenzyme. Polγ_A subunit is in white, Polγ_B1 in royal blue, and Polγ_B2 in sky blue. **b, c** Assembly of the editing complexes. DNA template strand (here and throughout) is shown in blue, the primer is in red, and the mismatched base is in yellow. **d** CryoEM density map of Polγ Mismatch Sensing complex (Structure I) at 2.8 Å. The Polγ_B homodimer is shown as a ribbon model (right panel) to reveal the Thumb subdomain (orange). The polymerase (*pol*) site is colored pink, and the exonuclease (*exo*) site is yellow. Major structural elements involved in proofreading - Wedge (teal), Guide loop (G-loop, smudge), and Sensor loop (purple)−are shown.

synthetic DNA scaffold with a primer having a terminal mismatched base connected via a nonhydrolyzable phosphorothioate bond (Complex ONE, Fig. 1b). In the second, we used a variant of Polγ, in which the catalytic residues in the *exo* site have been mutated[12] (D198A/E200A, Exo⁻ Polγ), and an RNA-DNA scaffold (Complex TWO, Fig. 1c). The primer in this complex has been extended by incorporating three cognate dGMP nucleotides and misincorporating one dGMP nucleotide against dTMP, generating a terminal mismatched base pair (Fig. 1c). Both complexes showed no significant exonuclease primer degradation within the time frame of the experiment (Supplementary Fig. 1a, b). In addition, to validate the data obtained using complex ONE, we prepared WT Polγ complex, in which ~60% of the mismatched DNA primer, containing hydrolyzable phosphodiester bond, has undergone proofreading (Complex THREE, Supplementary Fig. 1c).

The complexes were subjected to single-particle analysis using cryogenic-electron microscopy (CryoEM) (Supplementary Fig. 2–6, Supplementary Tables 1, 2). A series of high-resolution structures ranging from 2.6 to 3.1 Å provides a detailed account of the proofreading process (Figs. 1d and 2a, Supplementary Figs. 7 and 8). The CryoEM data revealed that Complex ONE and THREE were represented by five major 3D classes, while Complex TWO−by four major 3D classes, each with a conformation different from the one found in the catalytic Polγ complexes published previously[8,22] (Supplementary Figs. 2–6). Complexes ONE and THREE data sets produced a structure with a mismatched base in the *pol* site (Structure I, "Mismatch Sensing" complex), a structure with the mismatched base uncoupled from the *pol* site (Structure II, "Mismatch Uncoupling" complex), a structure

with Polγ during initial backward translocation toward the *exo* site (Structure VII, "Backtracking Initiation" complex), a structure with a mismatched base at the entrance of the exonucleolytic channel (Structure VIII, "Wedge Alignment" complex), and a structure with two single-stranded nucleotides of the primer located in the exonucleolytic channel (Structure IX, "Primer Separation" complex) (Fig. 2a, Supplementary Figs. 2, 4, and 7a). Analysis of the Complex TWO dataset resulted in structures representing the consequent forward translocation of Polγ relative to the conformation observed in Structure II by one base pair (bp) (Structure III, "Mismatch Locking" complex) and by two bp (Structures IV, V and VI, "Guide Loop Engagement" complex) (Fig. 2a, Supplementary Figs. 3 and 7b). Structures IV-VI reveal the same location of the mismatched base of the primer relative to the catalytic sites but show notable changes in protein conformation (Supplementary Fig. 7b). Overall, the ensemble of structures represents a stepwise progression of the proofreading process with a single-nucleotide resolution (Fig. 2a, Supplementary Figs. 7 and 8, Supplementary Videos 1 and 2). At the beginning of the proofreading process, the mismatched base is located in the *pol* site (Structure I); upon completion of the primer's translocation, this base is found ~35 Å away in the *exo* site (Structure IX) (Fig. 2b). The remaining structures represent the consecutive steps along the primer translocation pathway, which were assigned based on the proximity of the 3′ end of the primer to the *pol* or *exo* site (Fig. 2b). The non-overlapping conformational states of the Polγ-DNA complexes reveal that upon recognition of the mismatched base and its removal from the *pol* site (Structures I and II), Polγ translocates forward until the 3′ end of the primer is positioned at the entrance to the channel that leads to the *exo*

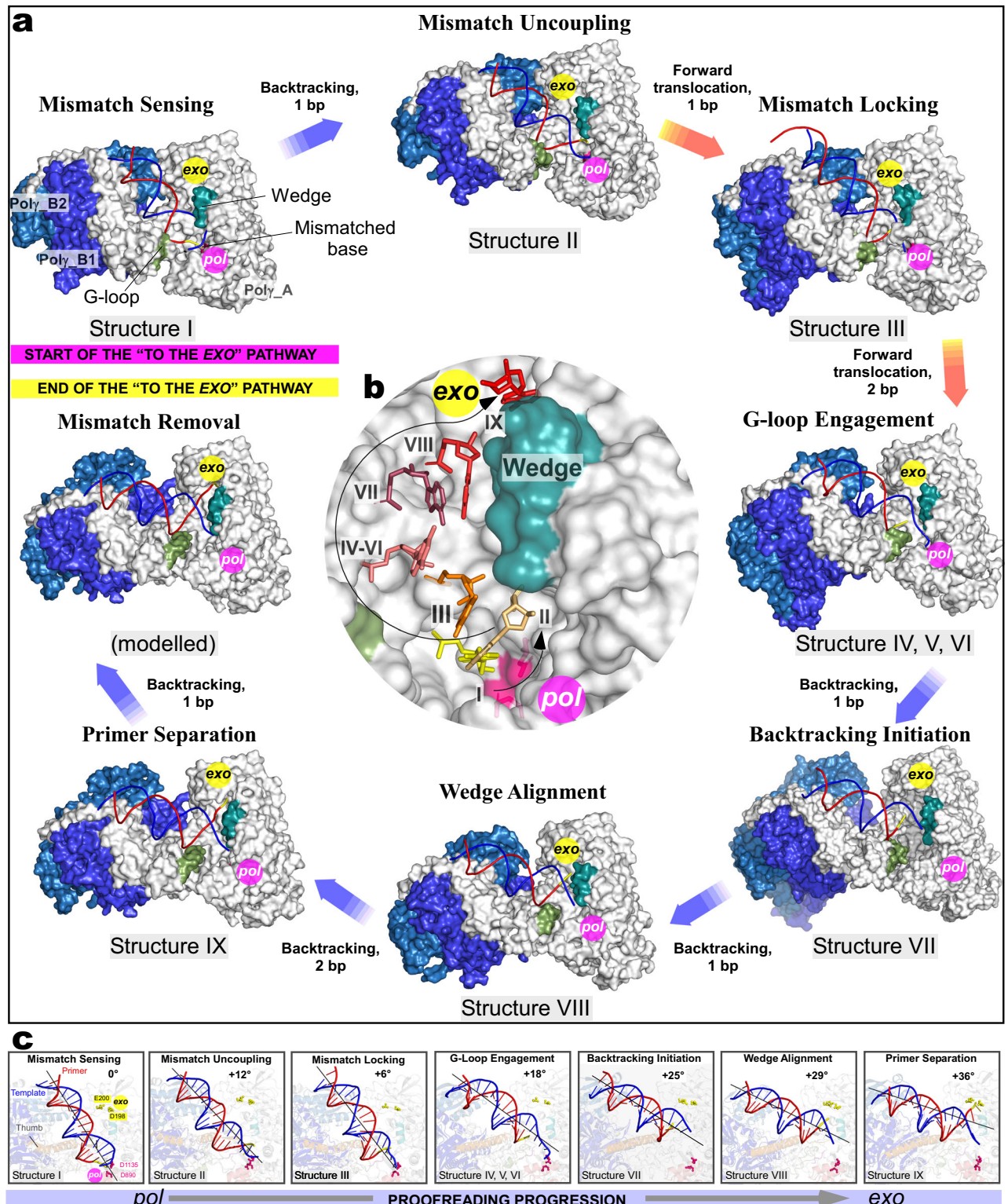

**Fig. 2 | CryoEM captures multiple steps along the DNA proofreading pathway.**
**a** Polɣ structures representing the major steps of the proofreading process. The complexes (Structures I–IV, VII, VIII, and IX, surface representation) are shown in the same orientation of their catalytic subunits. The Mismatch Removal complex is modeled using Structure IX to illustrate the final step of the proofreading. **b** The trajectory of the mismatched base during proofreading. The structures were aligned using the Polɣ_A subunits. The mismatched base of the primer is shown. **c** The change of DNA trajectory during proofreading. Structures are shown in the same orientation of their catalytic subunit. Change in DNA trajectory (degrees) is indicated relative to the DNA axis in the Mismatch-Sensing complex.

site (Structures III, IV, V, and VI, Fig. 2a). The transition from the Mismatch Uncoupling to the Mismatch Locking complex requires forward translocation of Polγ by one bp without major conformational changes (Supplementary Videos 3, 4). This suggests that Structures III-VI, obtained from analysis of the Complex TWO dataset, represent proofreading steps that follow the Mismatch Uncoupling but precede conformations captured in Structures VII-IX from Complex ONE (Fig. 2a, Supplementary Videos 1, 2). Upon positioning the 3′ end of the primer at the entrance of the exonuclease channel, Polγ translocates backward (Structure VII) to juxtapose the Wedge helix in the exonuclease domain next to the mismatched base pair (Structure VIII) and then to separate the 3′ end of the primer from the template and divert it towards the *exo* site in Structure IX (Fig. 2).

The final step of proofreading (Fig. 2a, "Mismatch Removal") and the corresponding structure have not been captured in our data sets due to experimental design since either the *exo* site (Complex TWO) or the 3′ terminus of the primer (Complex ONE) has been modified, affecting the affinity of the 3′ end of the primer towards the *exo* site. Nevertheless, the position of the primer termini poised for endonucleolytic reaction can be modeled based on the existing structure of the Klenow fragment of DNAP[4]. The progression of Polγ along the proofreading pathway is accompanied by ~40° change in the DNA axis relative to DNA in the catalytic structure (Fig. 2c) and by the relative motion of its subunits discussed below. Altogether, the observed conformations of Polγ represent consecutive steps in the process of translocation of the 3′ end of the primer from the *pol* site to the *exo* site (Fig. 2a, b), consistent with a non-dissociative (intramolecular) model of proofreading.

## Mismatch sensing and uncoupling from the *pol* site

The Complex ONE dataset revealed the initial stages of mismatch recognition (Fig. 3a), most likely because the phosphorothioate in the primer slows down the kinetics of DNAP translocation[13]. Structure I (Mismatch-Sensing complex, Fig. 3a) represents the open post-translocated conformation of the primer-template complex, in which the fingers subdomain (res 942–983) is disengaged from DNA. The mismatched base is found in the *pol* site at about the same distance from the catalytic aspartate residues as the cognate base[22] but tilted as compared to the canonical Watson-Crick base-pairing (Fig. 3c, d). The mismatched base pair is sensed via minor groove interactions with R853 and Q1102 residues, which are the functional analogs of R615 and Q797 in *Bacillus* DNAP I[23], and R429 and Q615 in T7 DNAP[9]. The interaction with the mismatch-sensing R853 residue is lost, but new hydrogen bonds are formed with the Q1102 residue of Polγ_A, altering the geometry of the terminal base pair (Fig. 3c). In addition, the Y955 residue of the O helix, which is also implicated in correct base-pairing sensing[24], partially occupies the substrate insertion site, impairing the extension of the mismatched primer (Fig. 3c).

Following mismatch recognition, the mismatched base is relocated ~15 Å away from the catalytic aspartate residues in the *pol* site in Structure II (Mismatch Uncoupling complex, Fig. 3b). Alignment of Structure I and II using the Polγ_B homodimer suggests that the upstream contacts of Polγ with the DNA are preserved and reveal the backward translocation of DNAP (Fig. 3e). This translocation is accompanied by a rigid body 9° rotation of the palm subdomain of the Polγ_A subunit relative to the thumb subdomain of Polγ_A, which remains bound to the upstream DNA (Supplementary Video 3). Movement of the Polγ_A palm subdomain around the axis nearly perpendicular to the DNA axis, accompanied by translocation of Polγ along the DNA, positions the 3′ end of the primer 15 Å away from the catalytic aspartate residues in the *pol* site (Fig. 3e). Alignment of Structure I and II using the conserved palm subdomain (res 815–910 and 1095–1236) suggests the movement of the "Sensor loop" (res 851–870), which harbors the mismatch-recognition residue R853 (Fig. 3f, g). Residues 861–864 of the Sensor loop are bulging away from

their position in the *pol* site of the Mismatch-Sensing complex, making its position incompatible with the primer trajectory in Structure I and contributing to the uncoupling of the mismatched base from the *pol* site (Fig. 3f, g).

## Forward translocation of Polγ during proofreading

Alignment of the Polγ_B homodimers in Structures II and III reveals the next step of proofreading (Mismatch Locking complex, Structure III), where forward translocation of Polγ by one base pair along the DNA axis is observed (Fig. 3h, Supplementary Video 4). Polγ moves as a rigid body, not affecting the relative trajectory of the DNA (RMSD 0.8 Å for 1335 Cα atoms). Six bases of the template strand of DNA are seen to neatly fit within the binding cavity of Polγ_A between the fingers and thumb subdomains in the Mismatch Locking complex (Fig. 4a, b). The first unpaired template base ($n+1$) is flipped out and partially stacks the mismatched base, assuming a "locking" conformation (Fig. 4a, b). The stacking interaction of the locking base with the mismatched base of the primer may prevent the premature fraying of the latter and its separation from the DNA template, ensuring timely entry to the exonuclease channel. The conformation of primer/template in Polγ observed in the Mismatch Locking complex is additionally stabilized by interactions with two conserved arginine residues (R337, R338) in the Wedge helix extension region, which becomes ordered in Structure III, and N803 and R807 residues in the thumb subdomain (Fig. 4a, b).

The subsequent step in the proofreading pathway involves forward translocation of Polγ along the DNA axis, rotation of the Polγ_A subunit relative Polγ_B homodimer, and refolding of the structural element termed the "Guide loop" (or G-loop, res 757–784), resulting in another distinct Polγ conformation (G-loop Engagement complex, Structure IV, Fig. 4c, d, Supplementary Video 5). As in Structure III, the DNA binding cavity of Polγ_A in the G-loop Engagement complex accommodates 6 bp of the single-stranded template DNA, which now runs nearly parallel to the Wedge helix (Fig. 4c). The $n+1$ base completes its rotation and is in a nearly perfect stacking conformation with the mismatched base (Fig. 4d), which likely limits any further forward translocation of Polγ. Compared to Structure III, the palm of Polγ_A gradually rotates relative to its thumb subdomain and Polγ_B2 by 13°–26° (Fig. 4c), assuming its most "open" conformation and changing the trajectory of DNA by 15° within the catalytic subunit in Structure VI (Fig. 2c). The distance between Polγ_B2 and the catalytic subunit increases to ~17 Å as measured by the position of the Arg 232 residue, which sits within the interacting distance with E394 residue in Polγ_B2 in Structures I–III (Fig. 4e). In contrast, the later steps (Structures IV-IX) show a significantly enlarged gap between them in the "open" conformation of Polγ (Fig. 4e). This relative movement of Polγ subunits appears critical for proofreading activity as it opens up the path for the primer toward the *exo* site, as has been proposed earlier[17].

The palm rotation of the Polγ_A subunit is accompanied by alterations in the thumb subdomain, which changes from a bent to a straight conformation (Fig. 4f, Supplementary Video 5). Alignment of the palm subdomains of Polγ_A in Structures IV and III shows the G-loop translation of ~14 Å and the fingers subdomain rotation by 15°, which pushes the 3′ terminus of the primer toward the *exo* site (Fig. 4b, d, g). A conserved residue in the G-loop, K768, is observed making a hydrogen bond with the phosphate backbone of the primer, moving ~13 Å from its position in Structure III. To probe the functional importance of the G-loop for proofreading activity of Polγ, we generated a deletion variant lacking residues 761–769 (ΔG-loop Polγ). We found that the enzyme's binding affinity (Supplementary Fig. 9a–c) and catalytic activity (Supplementary Fig. 9d) were not affected by this deletion. Because the base of the G-loop contributes to Polγ interactions with the DNA primer during primer extension, deletion of the G-loop residues results in a notable decrease in the rate of translocation (Fig. 4h). Importantly, ΔG-loop Polγ showed dramatically impaired

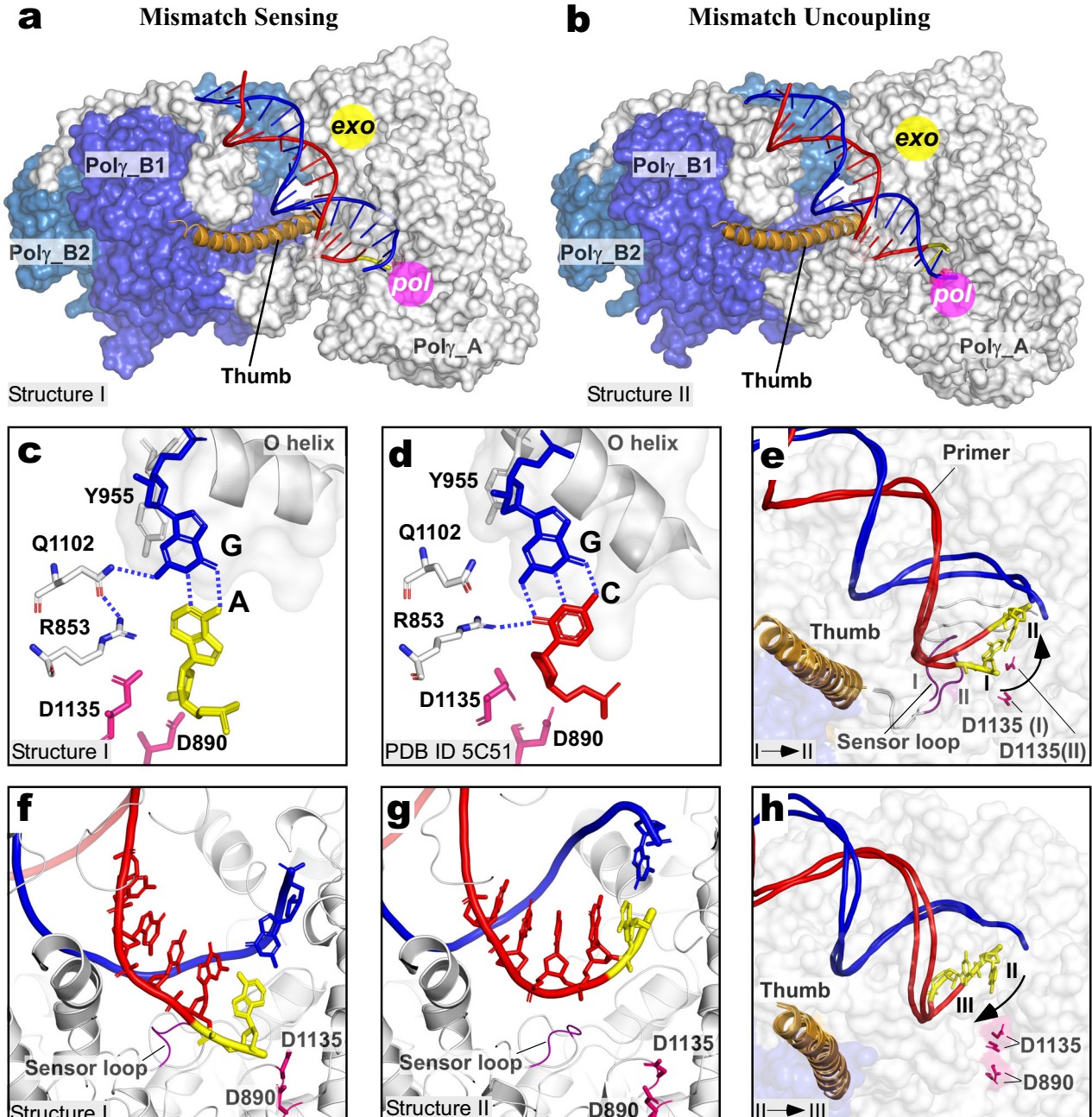

**Fig. 3 | Mismatched base sensing and uncoupling from the *pol* site.**
**a**, **b** Structures of the Mismatch-Sensing and Uncoupling complexes. The Polγ structures (surface representation) are shown in the same orientation of the Polγ_B homodimer. The thumb domain (res 439–476, orange) is shown as a ribbon representation. Parts of the N-terminal domain of Polγ_A (476–495, 592–623) and thumb (res 797–811) are omitted for clarity. **c**, **d** Close-up views of the *pol* active sites of the mismatched sensing (**c**) and catalytic (PDB ID 5C51, **d**) complexes. The complexes were shown in the same orientation of their conserved palm sub-domains, the catalytic *pol* site residues (D890/D1135) are in pink. **e** A close-up view

of DNA in the Mismatch-Sensing and Mismatch Uncoupling complexes. The structures were aligned using their Polγ_B homodimers. The mismatched bases are indicated in yellow, and the bulging part of the Sensor loop (res 861–864) is shown in purple. **f**, **g** Close-up views of the DNA binding cavity in Mismatched Sensing (**a**) and Mismatch Uncoupling (**b**) complexes. The complexes are shown in the same orientation of the conserved palm subdomain. **h** Polγ transition from Mismatch Uncoupling to Mismatch Locking complex. The structures were aligned using their Polγ_B homodimers. DNA strands and mismatched bases (yellow) are shown.

exonucleolytic activity on the mismatched scaffold, confirming the key role of the G-loop in proofreading (Fig. 4i). As mentioned above, Structures IV–VI show similar overall conformations (Fig. 4c). The most notable difference between these conformations is the gradual motion of the G-loop (Fig. 4g), accompanied by the straightening of the thumb and rotation of the palm subdomain of the Polγ_A subunit, suggesting an essential role of these elements in the proofreading process.

## Polγ backtracking and primer separation
The transition to the G-loop Engagement complex aligns the 3′ end of the mismatched primer with the entrance to the narrow channel leading to the *exo* site (Fig. 4c). Backward translocation of Polγ is now needed to separate the primer from the template strand and to deliver the mismatched base into the *exo* site (Fig. 5a–c). In the Backtracking Initiation complex, Polγ has translocated 1 bp backward as compared to the G-loop Engagement complex observed in Structure VI (Fig. 5a,

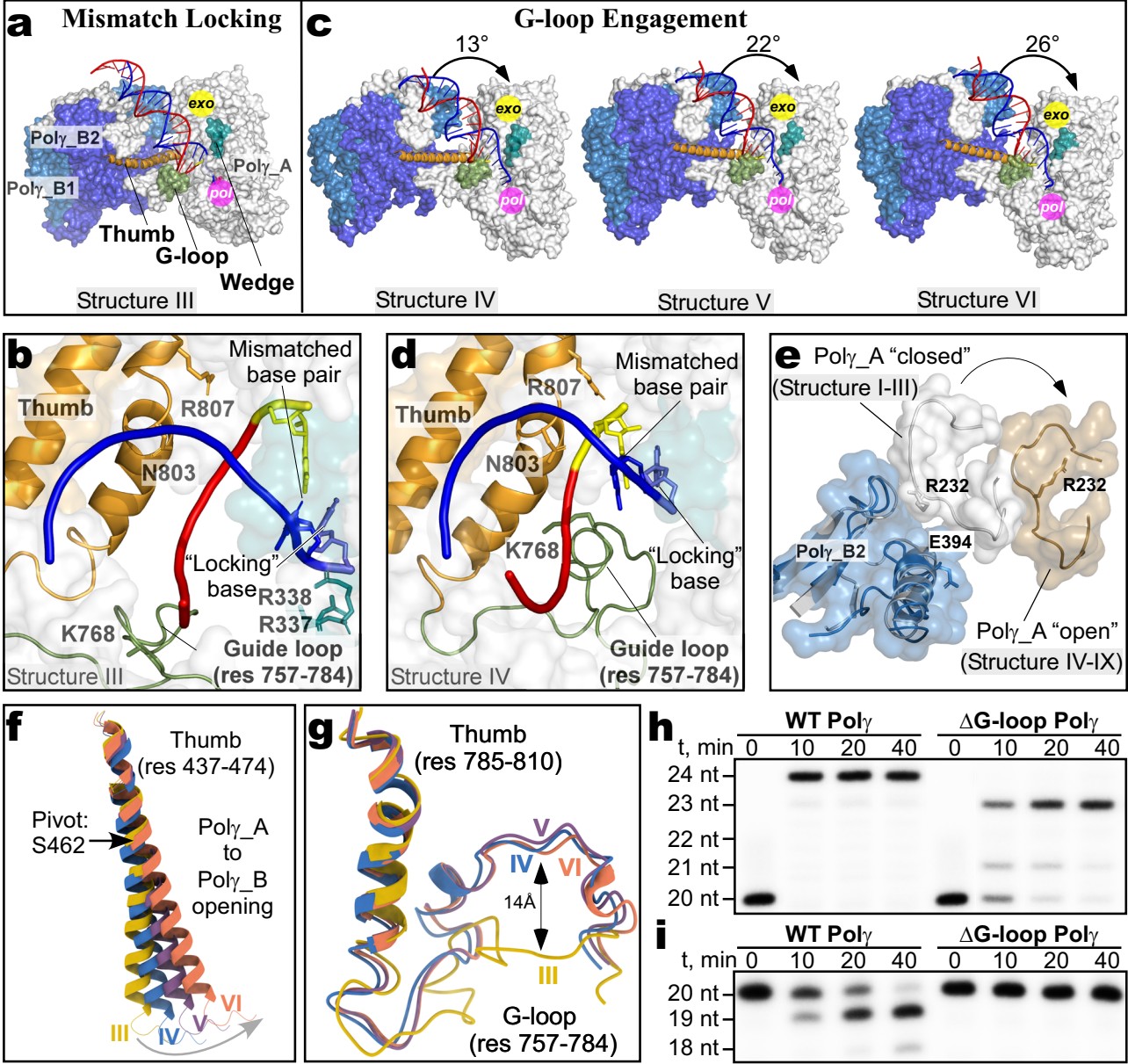

**Fig. 4 | Forward translocation of Polγ during proofreading. a** Structure of the Mismatch Locking complex. The Wedge helix (teal) and the G-loop (smudge) are shown. **b** Close-up views of the 3′-end of the DNA in the Mismatch Locking complex. The $n + 1$ "locking" DNA base is shown in magenta. **c** Structures of the Guide Loop Engagement complex (Structures IV–VI). The structures are shown in the same orientation of the Polγ_B homodimer as in the Mismatch Locking complex. Rotation of Polγ_A (degrees) is indicated relative to the Polγ_B homodimer in Structure III by the black arrow. **d** Close-up views of the 3′ end of the DNA in the Guide Loop Engagement complex. The complex is shown in the same orientation of the palm

subdomain of Polγ_A as in **b**. **e** Relative motion of Polγ_A and the distal Polγ_B2 subunits of Polγ during proofreading. Conformations in Structure I (closed) and Structure VI (open) are shown. **f** Close-up view of the long helix of the thumb subdomain in Mismatch Locking (III) and G-loop Engagement (IV–VI) structures showing the bending of the thumb during forward translocation. **g** Close-up view of the G-loop in Mismatch Locking (III) and G-loop Engagement (IV–VI) structures. **h** Primer extension assay using the G-loop deletion Polγ in the presence of dGTP. **i** The G-loop deletion variant of Polγ is inactive in the exonuclease assay. Gels in **h**, **i** are representative results from triplicate experiments.

d). The subsequent backward translocation places the Wedge helix atop the mismatched base pair, as observed in the next captured intermediate (Wedge Alignment complex, Structure VIII, Fig. 5b, d, e). Further backtracking by 2 bp is observed in the subsequent step of proofreading, which separates the primer from the template strand and positions the 3′ end in the *exo* site (The Primer Separation complex, Structure IX, Fig. 5c, f). Backtracking is defined as the process of backward translocation of polymerase along the DNA that results in a separation of the nascent RNA 3′ terminus from the catalytic site and is associated with the proofreading activity of RNA polymerase[25]. The process of backtracking is essential for many physiologically relevant

processes in bacteria and eukaryotes, such as transcription elongation, pausing, termination, fidelity, and genome instability[26]. The observed backward translocation of Polγ during proofreading suggests that backtracking may be a universal phenomenon for DNA and RNA polymerases.

In the Wedge Alignment complex, the invariant arginine residue (R309) is seen invading the space between the primer and template strand of DNA (Fig. 5e). Substitution of the R309 residue with alanine (R309A) significantly decreases Polγ exonuclease activity (Fig. 5g) while not affecting its translocation (Fig. 5h). The G-loop assumes its most extended conformation, while the N803, K806, and R807

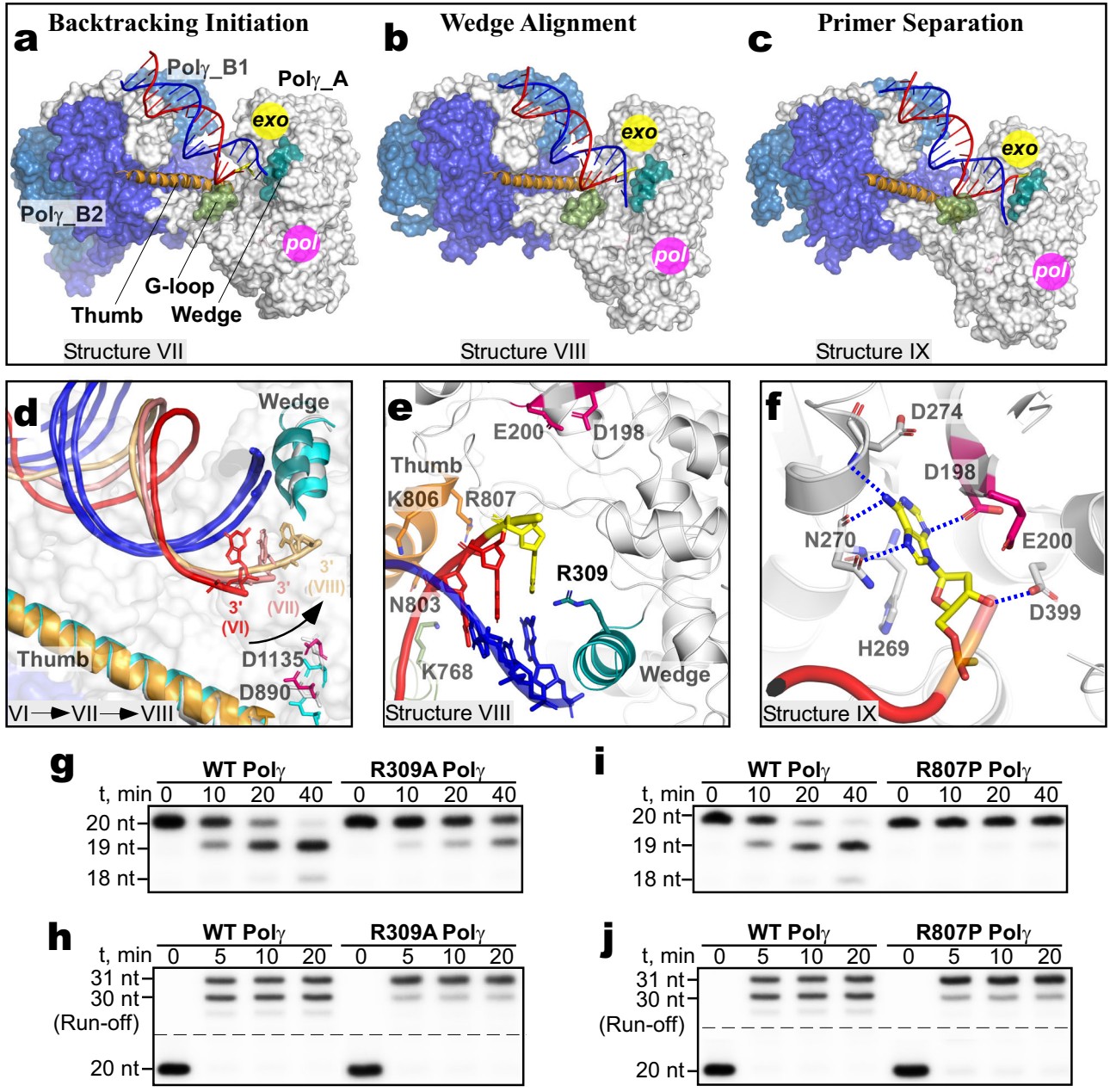

**Fig. 5 | Backward translocation of Polγ during proofreading. a** Structure of the Backtracking Initiation complex. **b** Structure of the Wedge Alignment complex. **c** Structure of the Primer Separation complex. The structure is shown in the same orientation of the Polγ_B homodimer as the Wedge Alignment complex in **b**. **d** Polγ transition from the Guide Loop Engagement (VI) to the Backtracking Initiation (VII) and Wedge Alignment complex (VIII). The structures were aligned using their Polγ_B homodimers. **e** Close-up view of the mismatched base pair at the entrance to the exonuclease channel in the Wedge Alignment complex (**b**). The *exo* catalytic residues (D198/E200) are in pink. **f** Close-up view of the 3′ end of the primer positioned in the *exo* site in the Primer Separation complex (**c**). **g, h** R309A Polγ variant is deficient in proofreading (**g**) but active in primer extension (**h**). **i, j** R807P Polγ variant is deficient in proofreading (**i**) but active in primer extension (**j**). Gels in **g**–**j** are representative results from triplicate experiments.

residues of the thumb subdomain additionally stabilize the DNA primer (Fig. 5e). Indeed, substituting the R807 residue with proline (R807P), a mutation found in patients with mitochondrial diseases[27–29], dramatically affects Polγ proofreading activity (Fig. 5i) while not affecting its translocation (Fig. 5j).

In the Primer Separation complex, the backtracking of Polγ pushes the Wedge helix against the mismatched base pair causing the primer to peel away from the duplex DNA and into the exonuclease channel (Fig. 5c, f). The backward motion is accompanied by a 12° rotation of the Polγ_A at a pivot located at the C-terminal part of the thumb

domain (res 476) (Supplementary Video 6). In the Primer Separation structure, two terminal nucleotides in the 3′ end of the primer are found in the exonuclease channel (Fig. 5c, f). The terminal base is inserted into the *exo* site and stabilized by hydrogen bonds with N270, D274, D198, and D399 residues (Fig. 5f). The proximity of the catalytic residues to the 3′ end of the primer suggests that the *exo* site of Polγ, similar to *E.coli* DNAP I[4], accommodates three nucleotides of a single-stranded DNA. Therefore, Polγ must backtrack one additional bp to position the phosphodiester bond between two terminal residues of the primer in the *exo* site in the Mismatch Removal complex (Fig. 2a).

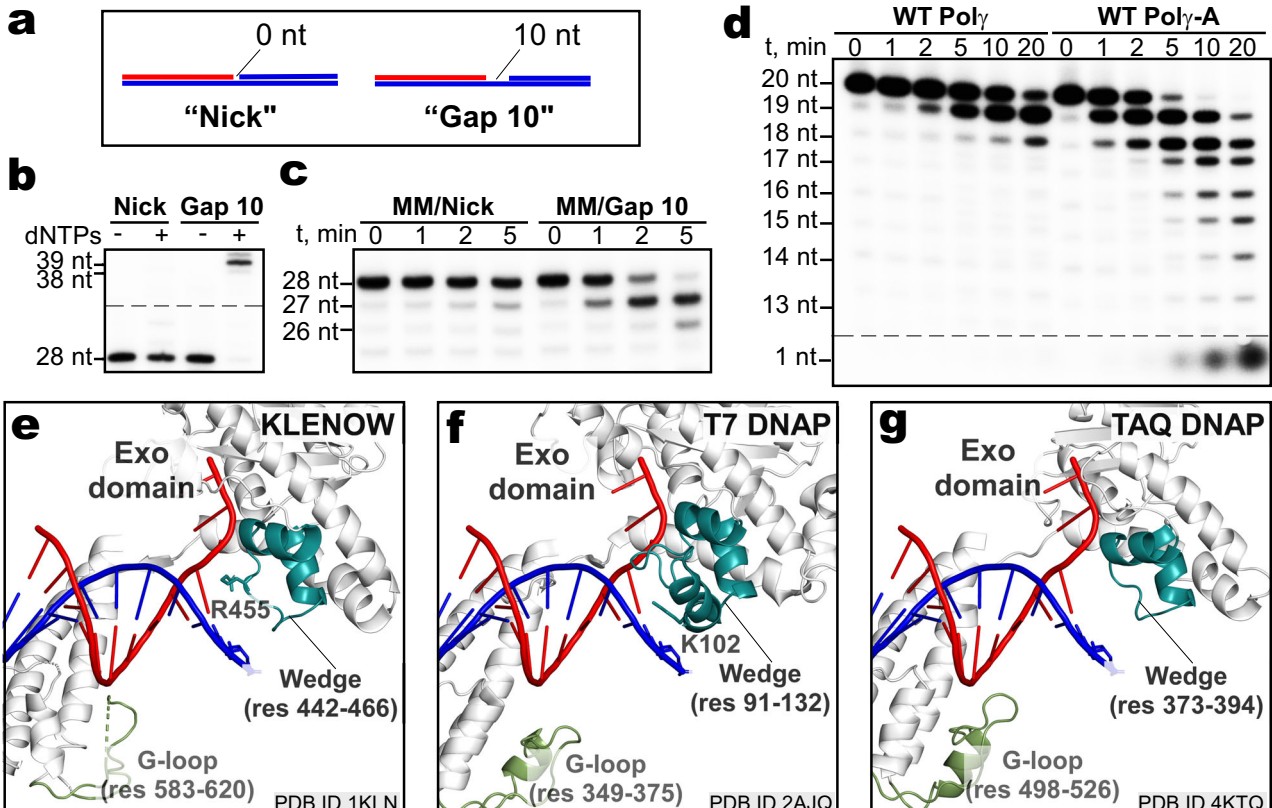

**Fig. 6 | The bolt-action mechanism of proofreading. a** Primer-template scaffolds with a nick or a 10-nt gap between the primer and downstream DNA duplex barrier used in (**b**, **c**). **b** Primer extension assays performed with WT Polγ in the presence of all dNTPs on scaffolds shown in (**a**). **c** Exonuclease assay performed with WT Polγ on scaffolds having a nick or 10-nt gap and a 3′ end mismatched (MM) base. **d** Exonuclease assays performed with WT Polγ and Polγ_A. **e**–**g** Structural elements involved in proofreading in Polγ are conserved in the Pol A family of DNAPs. Close-

up views of the major structural elements in Klenow (PDB 1KLN) (**e**), T7 DNAP (PDB 2AJQ) (**f**), and TAQ DNAP (PDB 4KTQ) (**g**) are shown. Structures are aligned with respect to their conserved palm subdomains. Primer-template DNA in the *exo* site (**f**, **g**) was modeled from the structure of the Klenow fragment (1KLN). Note that the existing T7 and TAQ structures do not reflect the proofreading step at which engagement of the G-loop is expected. Gels in **b**–**d** are representative results from triplicate experiments.

## The "bolt-action" mechanism of proofreading

The intermediate steps of the proofreading process are represented by distinct conformations of Polγ, which are stabilized by structural elements involved in the proofreading activity (Fig. 2a). These elements—the Sensor loop, the Guide loop, and the Wedge—contain highly conserved residues in mammalian species and are the hot spots for mutations that are associated with debilitating mitochondrial diseases (Supplementary Fig. 10). The prevailing point of view in the field of mitochondrial biology is that spontaneous replication errors are indeed responsible for the manifestation of mitochondrial diseases and premature aging[1,30,31]. Importantly, the link between Polγ proofreading deficiency and mitochondrial dysfunction was further established by the generation of a homozygous knock-in mouse model, termed "mtDNA-mutator" mice, and a genomically engineered fruit fly model, both expressing an exonuclease-deficient mutant of Polγ_A[32,33]. These studies have shown that Polγ proofreading deficiency led to the accumulation of mtDNA point mutations and deletions, displaying reduced lifespan and premature aging-related phenotypes in the mice and embryonic lethality in the fruit flies[32–34]. Additionally, others have identified and characterized exonuclease-deficient Polγ_A mutants that are implicated in disease[24]. Within the scope of this study, we have characterized two additional disease mutations, R309A and R807P, and have provided evidence that defects in structural elements required for Polγ proofreading activity result in proofreading errors, which in turn may be responsible for mtDNA mutations and, therefore, the onset of disease and aging.

The captured states of Polγ also suggest a "bolt-action" mechanism of the editing process. The "bolt-action" term refers to the iterative cycles of traversing the 3′ end of the DNA primer between the two catalytic sites, first from the *pol* site to *exo* site to eliminate the mismatched base, and then from the *exo* to *pol* site to return the corrected 3′ end of DNA (Fig. 2a, Supplementary Videos 1, 2). To demonstrate the importance of Polγ translocation for proofreading, we prevented the enzyme's ability to advance towards the downstream DNA region by introducing a non-template strand, as a barrier, into the scaffold constructs (Fig. 6a). The presence of the non-template strand did not affect Polγ binding affinity to DNA (Supplementary Fig. 11). In the absence of the replicative helicase TWINKLE, Polγ was not able to unwind the DNA and translocate forward on a nick-containing scaffold but incorporated dNTPs when loaded on a gap-10 scaffold (Fig. 6b). When scaffolds with the same topology but a mismatched primer were used, proofreading was observed only on the gap-10 scaffold (Fig. 6c), confirming that forward translocation is required for proofreading.

Since mismatch removal has been prevented by the scaffold design (Complex ONE) or the use of Exo⁻ Polγ (Complex TWO), we could not trap the return pathway of the corrected primer from the *exo* site to the *pol* site. Kinetic experiments suggest that this reverse pathway is much faster than delivering the mismatched base into the *exo* site[16]. Therefore, it is unclear if it involves the intermediate complexes described in this study. However, based on the observed changes in DNA trajectory, forward and backward translocations of Polγ are expected in order to reanneal the corrected primer to the DNA template and deliver its 3′ end back to the *pol* site (Supplementary Videos 1, 2).

As previously proposed, proofreading activity is associated with an equilibrium between DNAP's polymerization and proofreading modes[15,20]. This equilibrium can be shifted by altering catalytic properties of the *pol* or *exo* sites[35] and mutations affecting interactions of Polγ_A with Polγ_B homodimer[36–38]. Our findings suggest that during proofreading Polγ_B homodimer affects the motion of the thumb subdomain of Polγ, a structural element also involved in the enzyme's translocation along DNA during replication[22]. Indeed, in the absence of the Polγ_B homodimer, the catalytic subunit of Polγ exhibits strong exonucleolytic activity (Fig. 6d). However, it cannot efficiently discriminate between correct and mismatched bases, resulting in near-complete primer degradation, as evidenced by the accumulation of small cleavage products. In contrast, the holoenzyme accurately removes the mismatched base leaving the primer intact (Fig. 6d), in agreement with the proposed role of Polγ_B in proofreading and replication activities[19,39–43].

Why doesn't Polγ immediately cleave the next nucleotide upon removal of the mismatched base from the primer? Analysis of structures VIII, IX, and the modeled Mismatch Removal complex suggest that upon mismatch excision, Polγ must completely withdraw the primer from the *exo* site, allowing the NMP product to diffuse out of the narrow exonuclease channel. We speculate that at this point - the conformation observed in the Wedge Alignment Structure - the pathway for the corrected primer to the *pol* site would be favored over its return to the *exo* site. We propose that the Polγ_B homodimer stabilizes the Polγ_A exonuclease domain, thereby controlling the NMP diffusion mechanism and progressive exonucleolytic cleavage. In contrast, the exonuclease domain in Polγ_A might assume more relaxed conformation and allow faster NMP escape, promoting the processive primer degradation (Fig. 6d).

Our data suggest that the "bolt-action" mechanism of proofreading is likely conserved in Pol A family of DNAPs, which also includes proofreading *E.coli* DNA polymerase I, bacteriophage T7 DNAP, and proofreading-deficient TAQ DNAP, and human DNAPs Theta and Nu. All of these DNAPs, contain structural elements that our study demonstrates are critical for proofreading activity. Thus, in both the Klenow fragment and T7 DNAP, the Wedge helix harbors bulky amino acid residue—R455 and K102, respectively. These residues are found in a similar orientation to Polγ_A R309 residue and may serve an analogous function in primer separation (Fig. 6e, f). Indeed, mutations of these residues result in dramatic changes in the proofreading activity of these enzymes[44,45], in agreement with the Polγ_A R309 mutation phenotype demonstrated in this study. In contrast, the Wedge element in TAQ polymerase[46] does not contain any residues with the bulky side chain in the Wedge element (Fig. 6g), which likely contributes to the lack of proofreading in this enzyme[47]. The G-loop element is also present in DNAP I and T7 DNAP (Fig. 6e, g); however, its role in proofreading in these polymerases has not been verified by mutagenesis or structural methods.

Our finding of distinct intermediates along the proofreading pathways in a member of the Pol A family of polymerases raises the question of whether polymerases of other families can also employ the intramolecular mechanism of proofreading[21,48]. Since some of these polymerases have different domain organizations (such as Pol III[49]) or use distinct structural elements for primer separation (e.g., the beta-hairpin in T4 DNA polymerase of the Pol B family[16]), the proofreading pathway likely involves different, as compared to Polγ, intermediate complexes. Nevertheless, the "bolt-action" mechanism of proofreading described in this study may be a common strategy used by all processive DNA polymerases with proofreading activity.

## Methods

### Protein expression and purification
N-terminal histidine-tagged human Polγ_B (residues 26–485) was expressed and purified as described previously[50]. To express human Polγ_B, BL21-CodonPlus (DE3)-RIPL (Agilent) were transformed with the respective plasmid and grown at 37 °C in LB media until OD$_{600}$ reached 0.5 units. The proteins were induced by the addition of 0.15 mM IPTG for 18 h at 16 °C. Polγ_B was purified by affinity chromatography using Ni-NTA beads (Thermo Fisher Scientific), followed by affinity chromatography using a HiTrap heparin HP column (GE Healthcare). The heparin column was equilibrated in buffer A (40 mM Tris, pH 8.0, 300 mM NaCl, 5% Glycerol, 5 mM β-mercaptoethanol), and the protein was eluted by linear gradient 0–70% of buffer B (40 mM Tris, pH 8.0, 1.5 M NaCl, 5% Glycerol, 5 mM β-mercaptoethanol). Peak fractions were pooled, concentrated, and stored at −80 °C.

Variants of N-terminal histidine-tagged human Polγ_A (res 26–1239) were obtained by site-directed mutagenesis (QuikChange, Agilent). Polγ_A variants (WT, G-loop deletion res 761–769, R807P, and R309A) were expressed using SF9 cells and purified as previously described for Exo⁻ (D198A, E200A) with modifications[50]. Briefly, Polγ_A was purified by affinity chromatography using Ni-NTA beads (Thermo Fisher Scientific), followed by affinity chromatography using a TSKgel Heparin-5PW column (Tosoh Bioscience). The heparin column was equilibrated in buffer A (40 mM Tris-HCl pH 7.9, 150 mM NaCl, 5% Glycerol, 5 mM 2-mercaptoethanol), and Polγ_A was eluted by 0–70% linear gradient of buffer B (40 mM Tris-HCl pH 7.9, 1.5 M NaCl, 5% Glycerol, 5 mM 2-mercaptoethanol). Peak fractions eluted at 33 mS/cm were collected and analyzed using SDS-PAGE, concentrated, and stored at −80 °C.

Polγ holoenzyme complex was reconstituted by incubating Polγ_A fractions after Ni-NTA chromatography with a twofold molar excess of purified Polγ_B for 10 min at room temperature. Polγ was purified by affinity chromatography on a TSKgel Heparin-5PW column (Tosoh Bioscience), equilibrated in the buffer described above. Peak fractions eluted at 40 mS/cm were collected and analyzed using SDS-PAGE, concentrated, and stored at −80 °C.

### DNA and RNA oligonucleotides and scaffold preparation
Synthetic DNA oligonucleotides (IDT DNA) and synthetic RNA oligonucleotides (Dharmacon) were used. Phosphorothioate oligonucleotides were purchased as a racemic mixture of the two di-astereomers. Primers sequences (all 5′ to 3′): GAAGACAGTCTGCGGCGCG*A (DNA20sA, the asterisk denotes the position of phosphorothioate, Complex ONE), GAAGACAGUCUGCGGCGCGC (RNA20, Complex TWO), GAAGACAGTCTGCGGCGCGC (DNA20), CCAAGTCAGAAGACA GTCTGCGGCGCGC (DNA28), CCAAGTCAGAAGACAGTCTGCGGCGC GA (DNA28A), GAAGACAGTCTGCGGCGCGA (DNA20A), GGTACAACT TGACGACATAGCGTG (DNA24). Template strand sequences (5′ to 3′): ACACACGCGCGCCGCAGACTGTCTTC (DNA20TS, Complex ONE), GGTAGATCCCGCGCGCCGCAGACTGTCTTC (DNA20_3C_TS, Complex TWO), GGTAGATCCCACGCGCCGCAGACTGTCTTC (DNA20_3C_MM_ TS), CGGTCGAGTCACGACTCCGATTATGCGCGCCGCAGACTGTCTTC TGACTTGG (DNA28TS), CGGTCGAGTCACGACTCCGATTATCACGCT ATGTCGTCAAGTTGTACC (DNA24TS). Non-template strand sequences: TCGTGACTCGACCG (10nt gap_NT), ATAATCGGAGTCGTGACTCG ACCG (Nick_NT).

To anneal, the scaffolds were diluted in water, heated for 7 min at 95 °C, and cooled down (1 °C/min) for 1 h to 25 °C in a thermocycler.

### Primer extension and exonuclease assays
The primers were 5′-[³²P]-labeled using [γ-³²P]ATP (3000 Ci/mmol) and T4 Polynucleotide Kinase (NEB). The complexes of Polγ or Polγ_A (50 nM) with labeled primer-template scaffolds were assembled in a buffer containing 40 mM Tris-HCl pH 7.9, 60 mM NaCl, 10 mM MgCl$_2$, and 20 mM 2-mercaptoethanol in the presence of BSA (0.1 mg/ml) for 5 minutes at room temperature.

Primer extension was performed using DNA20/DNA30_3C_TS (Figs. 3h, 4g, i), "Nick" (DNA28/DNA28TS/Nick_NT, Fig. 6b), "Gap 10" (DNA28/DNA28TS/10nt gap_NT, Fig. 6b), or DNA20/DNA20TS

(Supplementary Fig. 9d) in the presence of 0.1 mM dNTPs or 1 mM dGTP, as indicated. Exonuclease assays were performed with DNA20/DNA20_MM_3C_TS (Figs. 3i, 4f, h, 6d), "MM/Nick" (DNA28A/DNA28TS/Nick_NT, Fig. 6c), or "MM/Gap 10" (DNA28A/DNA28TS/10nt gap_NT, Fig. 6c), all of which contained a single mismatch at the 3′ terminal end of the primer. All reactions were carried out for the indicated times at room temperature and stopped by the addition of an equal volume of 95% formamide/0.05 M EDTA. The products of the reaction were resolved by 20% PAGE containing 6 M Urea and visualized by autoradiography using PhosphorImager (GE Healthcare).

## Electrophoretic mobility shift assay (EMSA)
The 5′-Cy3-labeled variants of DNA24 and DNA28TS were obtained from IDT DNA (Supplementary Fig. 9a, 11a, b). To perform EMSA, complexes of Polγ/DNA were assembled in a buffer containing 40 mM Tris-HCl pH 7.9, 60 mM NaCl, 10 mM MgCl$_2$, 5% glycerol, 20 mM 2-mercaptoethanol in the presence of BSA (0.1 mg/ml), and incubated with 150 nM of 5′-Cy3-labeled scaffolds for 10 min at room temperature. The reactions were resolved in 0.5% agarose gels run in 0.5X TBE buffer for 30 minutes at 100 V at 4 °C. The products of the reactions were visualized using Bio-Rad ChemiDoc™ imager and quantified using Bio-Rad Image Lab™. For each reaction, the fraction of bound DNA was determined as the intensity of the free DNA divided by that in the 0 nM Polγ control (n = 3 independent experiments).

## Preparation of Polγ Complexes for CryoEM
To assemble Complex ONE, 2 μM WT Polγ was mixed with the DNA20*A/DNA20_TS scaffold at a 1:1.1 molar ratio in a buffer containing 10 mM Tris-HCl pH 7.9, 100 mM NaCl, 10 mM DTT, and 2 mM MgCl$_2$ and incubated for 5 minutes at room temperature prior to overnight dialysis at 4 °C in the same buffer.

Complex TWO (2 μM) was assembled using Exo⁻ Polγ (D198A, E200A) and the RNA20/DNA20 3C_TS scaffold at a 1:1 molar ratio in a buffer described above. Following incubation at room temperature for 5 minutes, dGTP was added to a final concentration of 0.1 mM, and the primer was extended for 2 minutes at room temperature, followed by dialysis.

To assemble Complex THREE, 2 μM WT Polγ was mixed with the DNA20A/DNA20_TS scaffold at a 1:1 molar ratio in a buffer containing 10 mM Tris-HCl pH 7.9, 100 mM NaCl, 10 mM DTT, and 2 mM MgCl$_2$ and incubated for 20 minutes at room temperature. The progression of mismatch cleavage was monitored using exonuclease assays described above.

The complexes were applied to negatively glow-discharged 300 mesh UltraAufoil −1.2/1.3 holey-gold grids (Quantifoil). Grids were blotted with ash-free Whatman® Grade 540 filter paper in a Vitrobot Mark IV (ThermoFisher Scientific) for 4–5 seconds at 4 °C and 95–100% humidity, then vitrified in liquid ethane. The sample quality and distribution were assessed using Glacios Transmission Electron Microscope equipped with a Falcon 4 direct electron detector.

## Single-particle data acquisition and image processing
Polγ data was collected at the Pacific Northwest Center for CryoEM (PNCC) using a Titan Krios transmission electron microscope (ThermoFisher Scientific), operated at 300 kV and equipped with a Bioquantum Energy Filter with a 20 eV slit width. Movies were collected using a Gatan K3 direct electron detector in super-resolution mode with a magnification of 105,000, corresponding to a pixel size of 0.413. A dose rate of 15.9–19.3 e⁻/s/physical pixel resulted in a total electron dose of 60–70 e⁻/Å², which was applied over 60–70 frames. Data was collected in SerialEM software with defocus values ranging from −0.5 to −2.0 μm.

Workflows for image processing of Complex ONE, Complex TWO, and Complex THREE are shown in Supplementary Figs. 2–4, respectively. The movie stacks collected for Complex ONE and Complex TWO were processed in CryoSPARC[51]. The super-resolution movies were frame aligned, motion corrected, gain normalized, dose-weighted, and binned twice with the patch motion correction module. Contrast transfer function (CTF) values were estimated using the patch CTF (CryoSPARC) or CTFFIND4[52]. Micrographs with ice, ethane contamination, and/or poor CTF fit resolution were discarded. A circular blob picker with dimensions of 80–130 Å was used to pick Polγ particles. Local resolution plots were obtained in CryoSPARC. Resolution values for the Polγ_A and Polγ_B subunits were computed in RELION3.0[53] with focused masks (Supplementary Table 1). The reported resolutions of the CryoEM maps are based on FSC 0.143 criterion[54]. The isotropy of the 3D reconstruction of Structures I-IX was estimated by 3DFSC server[55], as shown in Supplementary Figs. 5 and 6. The CryoEM density for the 3′ end of the primer in the exonuclease channel of Structure IX was improved with 3D classification in CryoSPARC 4.0.

## Model building and structure refinement
Polγ_A, Polγ_B1, and Polγ_B2 from the Polγ catalytic complex (PDB ID: 4ZTZ [https://doi.org/10.2210/pdb4ZTZ/pdb]) were docked into the respective CryoEM maps from Complex ONE and TWO. The template, primer, and mismatched base were placed into the CryoEM density maps. The terminal mismatched guanine bases in Complex ONE structures were modeled as nonhydrolyzable phosphorothioate bases, with 50% occupancy assigned to each stereoisomer. DNA-B and RNA-A restraints from Coot were used to fit the polynucleotide chains. Additional density in Structure II resolved up to 2.6 Å resolution allowed modeling of the loop regions in Polγ_B1 (res 137–161, 169–179), Polγ_B2 (137–179), and Polγ_A (336–340). A polyalanine model for Polγ_B2 docked in a low-pass filtered map was used for Structures IV and IX. The local density fit of the modeled sequence was improved over an iterative process of amino acid fitting in Coot 0.9.8.5[56], which alternated with real-space refinement in PHENIX[57]. Real-space refinement was carried out with secondary structure and Ramachandran restraints. Comprehensive model validation was carried out with PHENIX and the PDB validation server (https://validate-rcsb-2.wwpdb.org/) and is summarized in Supplementary Figs. 5 and 6 and Supplementary Table 2. Map-to-model Fourier Shell Correlation plots for the nine structures in Complexes ONE and TWO were obtained in PHENIX (Supplementary Figs. 5 and 6). Figures and Supplementary Videos were generated with PYMOL and ChimeraX[58].

## Statistics and reproducibility
Experiments presented in Figs. 4h, i, 5g–j, and 6b–d and Supplementary Figs. 1a–c, 9b, d, and 11a, b were repeated at least three times. The representative gel images are shown.

## Reporting summary
Further information on research design is available in the Nature Portfolio Reporting Summary linked to this article.

# Data availability
The CryoEM maps and atomic coordinates were deposited in the Electron Microscopy Data Bank (https://www.ebi.ac.uk/emdb) under accession codes EMD-29745, EMD-29746, EMD-41091, EMD-29747, EMD-29748, EMD-29749, EMD-29751, EMD-29752, and EMD-29750, and in the Protein Data Bank under accession codes 8G5I, 8G5J, 8T7E, 8G5K, 8G5L, 8G5M, 8G5O, 8G5P, and 8G5N. Previously published protein structure data used for analysis in this study are available in the Protein Data Bank (www.rcsb.org) under PDB ID: 5C51 and 4ZTZ (human mitochondrial DNAP Gamma), 1KLN (Klenow fragment of *E.coli* DNAP I), 2AJQ (T7 DNAP), 4KTQ (*Thermus aquaticus* DNAP). Source data are provided with this paper.

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

## Acknowledgements

We thank current and former members of the Temiakov laboratory. We are indebted to Sean Mulligan, Theo Humphreys, and Marcelo De Farias from PNCC and Tara Fox (NCI) for their expert technical assistance during data collection. We also thank Drs. William McAllister, Michael Anikin, Gino Cingolani, and Hauke Hillen for the critical reading of the manuscript and fruitful discussion. This work was supported by NIH Grant R35 GM131832 (D.T.). A portion of this research was supported by NIH grant U24GM129547 and performed at the PNCC at OHSU and accessed through EMSL (grid.436923.9), a DOE Office of Science User Facility sponsored by the Office of Biological and Environmental Research, and NIH grant S10 OD030457. This research was partly supported by the NCI's NationalCryo-EM Facility at the Frederick National Laboratory for Cancer Research under contract HSSN261200800001E.

## Author contributions

Experimental design and conceptualization: G.B., A.R.N., D.T. Protein preparation and biochemical experiments: G.B., A.S., V.O.S., A.Z.O. CryoEM Data acquisition: A.R.N., K.H. Single-particle CryoEM analysis and model building: A.R.N. Writing—original draft: D.T. Writing—review & editing: D.T., G.B., A.R.N., A.Z.O.

## Competing interests

The authors declare no competing interests.
