## [Peer Review File · Nature Communications]

Structural Basis for DNA ProofreadingReviewer #1 (Remarks to the Author):

During replication, DNA polymerase uses distinct active sites (polymerase (pol) and exonuclease (exo) site) for polymerization and proofreading activities. However, the mechanism by which DNA polymerases coordinate these two activities and translocate a mismatched primer from the polymerase (pol) to the exonuclease (exo) site and return to the polymerase (pol) site after proofreading cleavage remains elusive. In this manuscript, Temiakov and colleagues reported an ensemble of high-resolution structures representing human mitochondrial DNA polymerase Gamma captured during consecutive proofreading steps. Based on these intermediate structures, authors suggest a 'bolt-action' mechanism of proofreading based on iterative cycles of DNAP translocation without dissociation from the DNA, which transfers the primer between two catalytic sites. This elegant work greatly advances our understanding of proofreading mechanisms during DNA replication. Therefore, I support publication. I only have a couple of minor comments listed in below for clarification.

1. Page 3, line 1 Fig 1C. It is not clearly explain why authors use an RNA:DNA hybrid in Complex 2. Authors observed significant conformational changes between Structure II (Complex 1 using regular DNA duplex template/primer) and Structure III (complex 2 using RNA: DNA hybrid). How would authors rule out the possibility that these conformational changes are caused by the different conformations of regular template/primer DNA duplex vs RNA:DNA hybrid template/primer duplex (and subsequent rearrangement of nearby DNAP residues/motifs that interact with template/primer duplex)?

2. Page 5, line 9: shall be T7 DNAP. Ref 11 is for T7 DNAP instead of T7 RNAP.

Reviewer #2 (Remarks to the Author):

This report claims novel findings about how a polymerase shifts between the DNA synthesis site and the exo site. The paper is good, but I don't believe it is the first time that a Pol/Exo have been observed to be able to switch without enzyme dissociation. Certainly not the case in bacterial replisomes (eg they are highly processive on long substrates, yet still produce dNMPs, and thus assumed to be processive in switch between Pol and Exo sites while bound to their respective clamp. However, I am not an expert on the large variety of bacterial, archaeal and eukaryotic DNA polymerases that have been studied, but there certainly have been many previous structural and biochemical studies of the catalytic sites within various Pols. If the current work believes their work is novel, they should be sure to cite the many studies before theirs, that determine that the Pol comes off DNA for the Exo to function (which does happen for some Pols).

Importantly, the authors must do some biochemistry to prove the processivity of the Pol when it shifts between Pol and exo sites (ie add a trap, or a challenge template, or use of different amounts of enzyme/DNAs for example – and also to monitor dNMP production during the processive phase). Processivity cannot be assumed from static cryoEM structures. Also, the term for a Pol staying on DNA between these two modes, exo and pol, is "processive", not "bolt action". The term bolt action, is quite confusing – and suggests a gun, where the bolt both disposes of a spent cartridge and inserts a new cartridge, all within the same site. Thus, bolt action may not be what the authors are meaning since there are clearly two sites (exo and pol). I believe the term "processive" is the term the authors mean, but as I mentioned – this is not a novel finding, but might be for Pol gamma of this manuscript.

While the current study seems to be technically fine for structures, its conclusions require biochemical experiments, and in my opinion it lacks the substance that one might expect of a Nature Communications paper. I would suggest that it belongs in a more specialized journal.

I will only make a few minor suggestions, as the major ones are presented above:

-The various structures are interpreted to belong in a chronological order, although one can not be completely sure if this is the case. However, the authors logic flow is reasonable, and especially

considering the large amount of literature that already exists around these conclusions. But – the authors should acknowledge that they assume the chronological order of the structures, based on the extensive prior literature.

-CryoEM work only takes a subset of the total particle number to build a model. Therefore it cannot be clear that putative EM states are truly on the path of the enzyme mechanism or not. The authors should provide some level of statistics within the Results section as to why each intermediate is (or maybe not) directly on the reaction pathway.

-Fig. 1c does not show the mismatched bp that the text in the Results explains.

- Given that DNA Pols are very dynamic, and do not always follow the “expected” outcome when omitting a base, or dealing with thiolate links, it is not fully convincing that the Pol makes the movements noted in the report that uses non-natural linkages or Pol mutants. However, there is no other way to study these events, so it is fine by this reviewer. But the authors should note that these hugely dynamic machines can do things one would not expect when they are made to “stand still” because in my experience they never stand still. They find solutions around the most weird obstacle – somehow.

-While the modeling and logic flow suggest a backward movement of the Pol, it not fully convincing that these movements are on the track of the mismatch recognition path.

-Back tracking, as defined in this study, is not novel. It has long been thought that DNA Pols unwind a certain number of bp to correct a mismatch. So the correlation of backtracking in RNAP to DNAP may not be as direct or as novel as the authors suggest.

Regardless of these minor comments, the main comment is that these concepts are not novel – and the conclusions would still need biochemical “processivity” experiments to back them up.

Reviewer #3 (Remarks to the Author):

The manuscript by Buchel et al. provides the structural account of a proofreading reaction in a DNA polymerase (DNAP). The study relied on single-particle cryoEM to capture multiple intermediates of the proofreading process with single-nucleotide resolution to identify previously unknown structural elements in human mitochondrial DNA polymerase Pol gamma, as well as in more well-known polymerases of the Pol A family, such as Klenow fragment of E coli DNAP I and bacteriophage T7 DNAP. The role of particular domains in proofreading and the mechanism of proofreading is confirmed by well-designed biochemical experiments and mutagenesis. The study proposes a “bolt-action” mechanism of proofreading that is universal for at least the Pol A family of DNAP (and possibly for other families of processive high-fidelity DNA polymerases) and explains why mutations found in various regions of Pol gamma from patients with mitochondrial diseases affect the enzyme’s activity, which has implications for the development of therapeutic strategies.

The breadth of the cryoEM studies and the quality of the presented cryoEM maps are remarkable and illustrate the power of single-particle cryoEM to decipher dynamic processes at nearly atomic resolution. The movies are well-scripted and helpful in understanding the complexity of the proofreading process. Although the idea that DNAP relies on its exonuclease activity has been known for many years (since Tom Steitz’s seminal structures of Klenow fragment), the details of this process were not elucidated for any high-fidelity DNAP until now. It is interesting to see a familiar enzyme such as Klenow DNAP, which was thought to lack the structural elements needed to perform proofreading in a non-dissociative manner, possess such a dramatic conformational dynamic. To my knowledge, this study is also the first documented account of the backtracking process in DNAPs. The backtracking phenomenon was discovered over twenty years ago in RNA polymerases and has impacted our understanding of the molecular mechanisms of transcription

fidelity. The current study convincingly demonstrates the importance of backtracking for proofreading activity by DNAPs. Finally, the authors presented a wealth of structural information to support the proposed "bolt-action" mechanism without overwhelming the reader with unnecessary details. The manuscript is well-written and engaging.

Overall, I support the publication of this study pending some minor points:

1. The cryoEM studies clearly indicate the intramolecular nature of the proofreading pathway and assignment of the structures to a certain step of the process based on proximity of the 3' end of the primer to catalytic sites is logical. However, based on kinetic studies, it has been proposed that DNAP can dissociate from the primer/template and re-bind the primer in the exonuclease site. Can the authors completely exclude this based on their data?
2. A related paper published recently (Park J. et. al., NSMB 2023) provides several structures that represent the Pol gamma proofreading pathway. The authors should comment on how these structures relate to the structures presented in their manuscript (Structure I appears to represent the same initial step of mismatch sensing) and include this in a separate brief paragraph starting with the words "While this manuscript was in preparation...".
3. There are other Pol A family DNA polymerases in human cells. Do the authors suggest they also rely on the "bolt-action" mechanism of proofreading?
4. Many pathogenic mutations in Pol gamma are found in the regions identified in this study as critical for proofreading (Figure S10). The authors should explain these findings in a way to establish a direct connection between proofreading and mitochondrial diseases and aging, such as discussing previous genetic studies using the "mutator" mouse model and in flies (Trifunovic, et al Nature, 2004. 429(6990), Bratic et al, Nat Comm 2015).
5. The term backtracking has been associated with specific actions of RNA polymerase (cite PMID: 22726433) and has not yet been adopted in the DNA polymerase field. Therefore, some introductory definition of the term is justified here.

REVIEWER COMMENTS

We thank all reviewers for their constructive criticism that helped to improve the manuscript. Our point-by-point response is shown below in blue ink.

Reviewer #1 (Remarks to the Author):

During replication, DNA polymerase uses distinct active sites (polymerase (pol) and exonuclease (exo) site) for polymerization and proofreading activities. However, the mechanism by which DNA polymerases coordinate these two activities and translocate a mismatched primer from the polymerase (pol) to the exonuclease (exo) site and return to the polymerase (pol) site after proofreading cleavage remains elusive. In this manuscript, Temiakov and colleagues reported an ensemble of high-resolution structures representing human mitochondrial DNA polymerase Gamma captured during consecutive proofreading steps. Based on these intermediate structures, authors suggest a 'bolt-action' mechanism of proofreading based on iterative cycles of DNAP translocation without dissociation from the DNA, which transfers the primer between two catalytic sites. This elegant work greatly advances our understanding of proofreading mechanisms during DNA replication. Therefore, I support publication. I only have a couple of minor comments listed in below for clarification.

Response: We thank the reviewer for their kind words.

1. Page 3, line 1 Fig 1C. It is not clearly explain why authors use an RNA:DNA hybrid in Complex 2. Authors observed significant conformational changes between Structure II (Complex 1 using regular DNA duplex template/primer) and Structure III (complex 2 using RNA: DNA hybrid). How would authors rule out the possibility that these conformational changes are caused by the different conformations of regular template/primer DNA duplex vs RNA:DNA hybrid template/primer duplex (and subsequent rearrangement of nearby DNAP residues/motifs that interact with template/primer duplex)?

Response: The RNA:DNA hybrid is a natural substrate for Pol γ as this enzyme initiates replication by utilizing primers made by mitochondrial RNA polymerase. We used an RNA:DNA heteroduplex because we have found that Pol γ is more stable on RNA:DNA templates as compared to DNA:DNA templates. This finding has been shown experimentally and has been discussed in our manuscript, which is currently under review elsewhere (Buchel & Temiakov, 2023). Analysis of structures obtained using DNA:DNA and RNA:DNA hybrids indicates that Pol γ nicely adjusts to the wider grooves of the latter, and this adjustment does not involve the structural elements implicated in proofreading. The transition from Structure II (DNA-DNA) to Structure III (RNA-DNA,

(RMSD = 0.89 Å over 1338 C α atoms) does not involve rearrangements of Pol γ residues or motifs, but only a single nucleotide translocation event. Substantial changes in Pol γ conformation involving Thumb and G-loop occur during the transition from Structure III to IV and V, all of which include an RNA:DNA primer-template. We therefore believe that the RNA:DNA hybrid *per se* is not the cause of the observed conformational changes.

2. Page 5, line 9: shall be T7 DNAP. Ref 11 is for T7 DNAP instead of T7 RNAP.

Response: We thank the reviewer for pointing this out. We have corrected this typo in the manuscript.

Reviewer #2 (Remarks to the Author):

This report claims novel findings about how a polymerase shifts between the DNA synthesis site and the exo site. The paper is good, but I don't believe it is the first time that a Pol/Exo have been observed to be able to switch without enzyme dissociation. Certainly not the case in bacterial replisomes (eg they are highly processive on long substrates, yet still produce dNMPs, and thus assumed to be processive in switch between Pol and Exo sites while bound to their respective clamp).

Response. In the manuscript, we never claimed to be the first to find that the Pol A family of DNAPs can proofread DNA without dissociation. The main point of this work is that we reveal the molecular mechanism by which the 3'-end is transferred between the *pol* and *exo* sites. The Introduction section thoroughly discusses references related to the existing intra- and intermolecular modes of proofreading. To emphasize this further, we have included the following explanation into the revised manuscript to help the reader distinguish between the proofreading process and processivity as follows:

"While there is a general agreement in the field that Pol A enzymes are processive and can proofread DNA without dissociating, the lack of understanding of how editing can be achieved without engagement of dedicated structural elements in DNAP has persisted until now."

However, I am not an expert on the large variety of bacterial, archaeal and eukaryotic DNA polymerases that have been studied, but there certainly have been many many previous structural and biochemical studies of the catalytic sites within various Pols. If the current work believes their work is novel, they should be sure to cite the many studies before theirs, that determine that the Pol comes off DNA for the Exo to function (which does happen for some Pols).

Response: The reviewer is correct: over 1,700 structures deposited to the Protein Data Bank represent apo forms of DNAPs and DNAPs in the catalytic mode. However, this study captured eight key non-catalytic structures of a DNA polymerase along the proofreading pathway. None of these non-catalytic states were previously reported for any known DNA polymerase.

Importantly, the authors must do some biochemistry to prove the processivity of the Pol when it shifts between Pol and exo sites (ie add a trap, or a challenge template, or use of different amounts of enzyme/DNAs for example – and also to monitor dNMP production during the processive phase). Processivity cannot be assumed from static cryoEM structures.

Response: The processivity of Pol γ , as well as other Pol A family DNAPs, has already been addressed previously using biochemical experiments (including trap assays), and the relevant studies have been extensively cited in the manuscript – Longley *et al.*, *Biochemistry*, 1998 (Ref 14); Johnson & Johnson, *JBC*, 2001 (Ref 19); Dangerfield & Johnson, *JBC*, 2022 (Ref 20). However, neither of these studies revealed the mechanism of proofreading, which remained unknown for nearly 30 years, as the controversy about the possible dissociation of the polymerase persisted. For example, a 1989 study by C. Joyce on *E. coli* DNA polymerase stated in the abstract, “There is nothing in the protein structure or the reaction mechanism that dictates a particular means of moving the DNA substrate.”

Our study reveals, for the first time, the structural elements required for the proofreading activity and the previously unknown path for the switching of the 3' end of the primer between the two catalytic sites and explains why this process can be processive in the Pol A family of DNAPs. The documented dynamic changes observed in polymerase and in the topology of nucleic acids within the proofreading complex constitute the manuscript's main point. Our conclusions on processivity are based not on a single (“static”) CryoEM structure, but on a revealed single-nucleotide resolution pathway represented by nine intermediates.

Also, the term for a Pol staying on DNA between these two modes, exo and pol, is “processive”, not “bolt action”. The term bolt action, is quite confusing – and suggests a gun, where the bolt both disposes of a spent cartridge and inserts a new cartridge, all within the same site. Thus, bolt action may not be what the authors are meaning since there are clearly two sites (exo and pol). I believe the term “processive” is the term the authors mean, but as I mentioned – this is not a novel finding, but might be for Pol gamma of this manuscript.

While the current study seems to be technically fine for structures, its conclusions require biochemical experiments, and in my opinion it lacks the substance that one might expect of a Nature Communications paper. I would suggest that it belongs in a more specialized journal.

Response: The reviewer appears to have missed the main point of the manuscript. The processivity of the Pol A family of DNAPs during proofreading is intuitive and has been proposed previously. This study provides the next level of understanding of an important biological process that was discovered many years ago– correction of errors during DNA replication by the inherent exonucleolytic activity of DNAP. This includes deciphering what elements in DNAP are responsible for recognition of the mismatched base and its removal from the active site, separating the 3' end of the primer from the template strand, and delivering the mismatched base into the exonuclease site. The entire proofreading pathway consists of multiple conformational states of polymerase stabilized by the newly established protein-DNA interactions. These states have never been identified or predicted, until now. For example, the processes of translation of RNA to protein (<https://pubmed.ncbi.nlm.nih.gov/36171285/>), utilization of proteins by a proteasome (<https://pubmed.ncbi.nlm.nih.gov/30479383/>), or CRISPR-Cas9 function (<https://pubmed.ncbi.nlm.nih.gov/35236982/>) has also been known for quite some time. It is the dynamics of these processes and the discovery of the mechanistic details behind them by utilizing CryoEM that delivers new depth to our understanding and merits publication of these studies in high-impact journals.

We used the term “bolt-action,” referring to the mechanism of certain rifles, to highlight the iterative cycles of forward and backward translocation of polymerase along DNA during the proofreading process. We agree that the term “bolt-action” should be explained more thoroughly, and the following explanation has been incorporated into the revised manuscript as follows:

“The “bolt-action” term refers to the iterative cycles of traversing the 3' end of the DNA primer between the two catalytic sites, first from the pol site to exo site to eliminate the mismatched base, and then from the exo to pol site to return the corrected 3' end of DNA.”

We also elected to change the title of the manuscript to de-emphasize the mechanistic term used for the proofreading process. The new title now reads: *“Structural Basis for DNA Proofreading.”*

I will only make a few minor suggestions, as the major ones are presented above:

-The various structures are interpreted to belong in a chronological order, although one can not be completely sure if this is the case. However, the authors logic flow is reasonable, and especially considering the large amount of literature that already exists around these conclusions. But – the authors should acknowledge that they assume the chronological order of the structures, based on the extensive prior literature.

Response: Unlike what the reviewer is suggesting here, there is no prior “extensive” literature available that has identified the path taken by the mismatched base in order to be transported from the *pol* to *exo* site. Therefore the sequence of events involved in the 3' end transfer is not based on prior observation but is original to this work. Moreover, some studies have hypothesized that proofreading requires only a single backtracking step, i.e. DNAP may translocate backward in order to separate the primer from the template strand. However, our study demonstrates that it was an incorrect assumption, as DNAP must move forward to position the strand separating element (Wedge helix) in close proximity to the mismatched base pair. The chronological ordering of our complexes is based on the distance of the 3' end of the primer from the *pol* or *exo* site, which is most logical for understanding the proofreading process. This order is further corroborated by the gradual changes in Poly structure during the proofreading process, such as repositioning of the Sensor loop (Fig 3e), opening of the exonuclease channel by the movement of Poly_B relative to Poly_A (Fig 4e), movement of the G-loop and Thumb subdomain (Fig 4g,f), changes in DNA topology, such as “locking” of the DNA base (Fig 4b) and changes in the DNA trajectory (Fig 2c).

-CryoEM work only takes a subset of the total particle number to build a model. Therefore it cannot be clear that putative EM states are truly on the path of the enzyme mechanism or not. The authors should provide some level of statistics within the Results section as to why each intermediate is (or maybe not) directly on the reaction pathway.

Response: Initial CryoEM datasets contain particles damaged during blotting, vitrification, and high electron beam exposure, and therefore multiple rounds of unbiased “clean up” are required to remove them and arrive at high-quality images used for 3D reconstruction and model building. Once the particle cleaning is achieved, all selected particles are used to build 3D models. An exhaustive statistical analysis has been shown in Extended Data Figures 2-6, Tables 1-2, and PDB Validation Reports 1-9. Each 3D class revealed by SPA is represented by a very large number of particles (from 0.4 to 1.2 million), contributing to the high quality of our structures. Moreover, the individual 3D classes, for example, Structures III, IV, V and VI within Complex TWO, account for 29, 20, 26, and 25% of total particle number, indicating that the major conformations observed represent the on-path intermediates.

-Fig. 1c does not show the mismatched bp that the text in the Results explains.

Response: In the experiment shown in Fig 1c, Poly misincorporates dGMP against dTMP, creating a G/T mismatch, which is highlighted in yellow. This experiment is discussed in the Result section as well.

- Given that DNA Pols are very dynamic, and do not always follow the “expected” outcome when omitting a base, or dealing with thiolate links, it is not fully convincing that the Pol makes the movements noted in the report that uses non-natural linkages or Pol mutants. However, there is no other way to study these events, so it is fine by this reviewer.

Response: The structures obtained using the natural progressing proofreading reaction in Complex THREE and the “trapped” reaction with non-hydrolyzable mismatched base in Complex ONE, are exactly the same (RMSD <0.8Å), eliminating the concerns that the presence of a non-natural linkage may have created an aberrant conformation of the complex.

But the authors should note that these hugely dynamic machines can do things one would not expect when they are made to “stand still” because in my experience they never stand still. They find solutions around the most weird obstacle – somehow.

Response: The reviewer is correct – polymerases never stand still, even when “idle”. This is the key difference between X-ray crystallography, which allows capturing a single conformation of the enzyme within the crystal lattice, and CryoEM, which provides an opportunity to obtain all possible conformations within the same dataset.

-While the modeling and logic flow suggest a backward movement of the Pol, it not fully convincing that these movements are on the track of the mismatch recognition path.

Response: Our structural data suggest that backtracking is needed to align the mismatched base with key structural elements (G-loop and the Wedge helix), which are required to reposition the mismatched base into the exonuclease channel and separate the primer from the template strand. These unique conformations have not been observed in published Pol γ catalytic states, indicating that they represent intermediates of the proofreading process.

-Back tracking, as defined in this study, is not novel. It has long been thought that DNA Pols unwind a certain number of bp to correct a mismatch. So the correlation of backtracking in RNAP to DNAP may not be as direct or as novel as the authors suggest.

Response: Backward translocation of DNA Polymerases to correct a mismatch has been suggested, but not proven by any previous biochemical or structural studies, and as indicated by Reviewer 3, has not been

adopted in the DNA polymerase field. To the best of our knowledge, our study is the first documented record of backtracking in a DNAP.

Regardless of these minor comments, the main comment is that these concepts are not novel – and the conclusions would still need biochemical “processivity” experiments to back them up.

Response: We have addressed this comment in detail above.

Reviewer #3 (Remarks to the Author):

The manuscript by Buchel et al. provides the structural account of a proofreading reaction in a DNA polymerase (DNAP). The study relied on single-particle cryoEM to capture multiple intermediates of the proofreading process with single-nucleotide resolution to identify previously unknown structural elements in human mitochondrial DNA polymerase Pol gamma, as well as in more well-known polymerases of the Pol A family, such as Klenow fragment of E coli DNAP I and bacteriophage T7 DNAP. The role of particular domains in proofreading and the mechanism of proofreading is confirmed by well-designed biochemical experiments and mutagenesis. The study proposes a “bolt-action” mechanism of proofreading that is universal for at least the Pol A family of DNAP (and possibly for other families of processive high-fidelity DNA polymerases) and explains why mutations found in various regions of Pol gamma from patients with mitochondrial diseases affect the enzyme’s activity, which has implications for the development of therapeutic strategies.

The breadth of the cryoEM studies and the quality of the presented cryoEM maps are remarkable and illustrate the power of single-particle cryoEM to decipher dynamic processes at nearly atomic resolution. The movies are well-scripted and helpful in understanding the complexity of the proofreading process. Although the idea that DNAP relies on its exonuclease activity has been known for many years (since Tom Steitz’s seminal structures of Klenow fragment), the details of this process were not elucidated for any high-fidelity DNAP until now. It is interesting to see a familiar enzyme such as Klenow DNAP, which was thought to lack the structural elements needed to perform proofreading in a non-dissociative manner, possess such a dramatic conformational dynamic. To my knowledge, this study is also the first documented account of the backtracking process in DNAPs. The backtracking phenomenon was discovered over twenty years ago in RNA polymerases and has impacted our understanding of the molecular mechanisms of transcription fidelity. The current study convincingly demonstrates the importance of backtracking for proofreading activity by DNAPs. Finally, the

authors presented a wealth of structural information to support the proposed “bolt-action” mechanism without overwhelming the reader with unnecessary details. The manuscript is well-written and engaging.

Response: We thank the reviewer for their appreciation of our findings.

Overall, I support the publication of this study pending some minor points:

1. The cryoEM studies clearly indicate the intramolecular nature of the proofreading pathway and assignment of the structures to a certain step of the process based on proximity of the 3' end of the primer to catalytic sites is logical. However, based on kinetic studies, it has been proposed that DNAP can dissociate from the primer/template and re-bind the primer in the exonuclease site. Can the authors completely exclude this based on their data?

Response: Indeed, earlier kinetic studies using DNAP I suggested that both dissociative and non-dissociative modes of proofreading exist. More recent kinetic studies [Dangerfield & Johnson, *JBC*, 2022 (Ref 20)], however, indicate that DNAP does not dissociate from the template during proofreading, consistent with our observation of multiple intermediates. Importantly, in our dataset, we did not detect any significant presence of free Pol γ , suggesting that all intermediates of the proofreading pathway are stable and dissociation of Pol γ during editing is highly unlikely.

2. A related paper published recently (Park J. et. al., NSMB 2023) provides several structures that represent the Pol gamma proofreading pathway. The authors should comment on how these structures relate to the structures presented in their manuscript (Structure I appears to represent the same initial step of mismatch sensing) and include this in a separate brief paragraph starting with the words "While this manuscript was in preparation...".

Response: The recent study mentioned by the reviewer presents three structures of Pol γ in complex with DNA having a mismatched base at the 3' end of the primer. Two of these structures correspond to the initial stages of proofreading, Mismatch Sensing and Mismatch Uncoupling, while the other is the endpoint of the proofreading process (Mismatch Removal, Structure X). However, their exact placement in the proofreading pathway and physiological relevance are difficult to assess due to the presence of calcium ions that do not support the proofreading activity of the enzyme and may have affected the local conformations of DNA and Pol γ . For this reason, we elected not to comment on the work by Park et al., unless the reviewer insists on doing so.

3. There are other Pol A family DNA polymerases in human cells. Do the authors suggest they also rely on the “bolt-action” mechanism of proofreading?

Response: Our analysis indicates that all members of the Pol A family of DNAPs contain structural elements implicated in proofreading. However, human DNA repair polymerases Theta and Nu, similar to TAQ DNAP, do not possess proofreading activity, likely due to changes in their exonuclease sites and/or other structural elements. We indicated this in the following sentence:

“Our data suggest that the “bolt-action” mechanism of proofreading is likely conserved in Pol A family of DNAPs, which also includes proofreading E.coli DNA polymerase I, bacteriophage T7 DNAP, and proofreading-deficient TAQ DNAP, and human DNAPs Theta and Nu.”

4. Many pathogenic mutations in Pol gamma are found in the regions identified in this study as critical for proofreading (Figure S10). The authors should explain these findings in a way to establish a direct connection between proofreading and mitochondrial diseases and aging, such as discussing previous genetic studies using the "mutator" mouse model and in flies (Trifunovic, et al Nature, 2004. 429(6990), Bratic et al, Nat Comm 2015).

Response: We thank the reviewer for bringing up this very important point. We have discussed the direct connection between proofreading and pathogenesis in our revised manuscript:

“The prevailing point of view in the field of mitochondrial biology is that spontaneous replication errors are indeed responsible for the manifestation of mitochondrial diseases and premature aging (Khrapko et al, PNAS, 1997; Zheng et al., Mutat Res, 2006; Lujan et al., Genome Biol, 2020). Importantly, the link between Poly proofreading deficiency and mitochondrial dysfunction was further established by the generation of a homozygous knock-in mouse model, termed “mtDNA-mutator” mice, and a genomically-engineered fruit fly model, both expressing an exonuclease-deficient mutant of Poly_A (Trifunovic et al., Nature, 2004 & Bratic et al., Nat Comm, 2015). These studies have shown that Pol γ proofreading deficiency led to the accumulation of mtDNA point mutations and deletions, displaying reduced lifespan and premature aging-related phenotypes in the mice and embryonic lethality in the fruit flies (Trifunovic et al., Nature, 2004; Vermulst et al., Nat Gen, 2008; Bratic et al., Nat Comm, 2015). Additionally, others have identified and characterized exonuclease-deficient Poly_A mutants that are implicated in disease (Ponamarev et al., JBC, 2002). Within the scope of this study, we have characterized two additional disease mutations, R309A and R807P, and have provided evidence that defects in structural elements required for Poly proofreading activity result in proofreading errors, which in turn may be responsible for mtDNA mutations and therefore, the onset of disease and aging.”

5. The term backtracking has been associated with specific actions of RNA polymerase (cite PMID: 22726433) and has not yet been adopted in the DNA polymerase field. Therefore, some introductory definition of the term is justified here.

Response: We thank the reviewer for pointing this out. We have included more explanation of the term backtracking and cited the relevant paper in the revised manuscript:

“Backtracking is defined as the process of backward translocation of polymerase along the DNA that results in a separation of the nascent RNA 3’ terminus from the catalytic site and is associated with the proofreading activity of RNA polymerase²⁵. The process of backtracking is essential for many physiologically relevant processes in bacteria and eukaryotes, such as transcription elongation, pausing, termination, fidelity, and genome instability²⁶.”

Reviewer #1 (Remarks to the Author):

The authors had addressed all my concerns in their revision. Therefore, I support for publication.

Reviewer #2 (Remarks to the Author):

The revised manuscript is now acceptable by me.

Reviewer #3 (Remarks to the Author):

The authors have successfully addressed all of my concerns as well as those raised by other reviewers. I am of the opinion that the revised manuscript is now in an acceptable state for publication.